# How human runners regulate footsteps on uneven terrain

**Nihav Dhawale[1,2], Madhusudhan Venkadesan[1]\***

[1]Department of Mechanical Engineering and Materials Science, Yale University, New Haven, United States; [2]National Centre for Biological Sciences, Tata Institute of Fundamental Research, Mumbai, India

**Abstract** Running stably on uneven natural terrain takes skillful control and was critical for human evolution. Even as runners circumnavigate hazardous obstacles such as steep drops, they must contend with uneven ground that is gentler but still destabilizing. We do not know how footsteps are guided based on the uneven topography of the ground and how those choices influence stability. Therefore, we studied human runners on trail-like undulating uneven terrain and measured their energetics, kinematics, ground forces, and stepping patterns. We find that runners do not selectively step on more level ground areas. Instead, the body's mechanical response, mediated by the control of leg compliance, helps maintain stability without requiring precise regulation of footsteps. Furthermore, their overall kinematics and energy consumption on uneven terrain showed little change from flat ground. These findings may explain how runners remain stable on natural terrain while devoting attention to tasks besides guiding footsteps.

## Editor's evaluation

This paper presents fundamental evidence for the control mechanisms used by running humans to maintain stability while running on naturalistically uneven terrain. The authors use a creative and compelling combination of experiments and modeling to analyze running on terrain with mildly stochastic undulating roughness, a condition that resembles natural terrain conditions, such as trail running. The findings suggest that humans use open-loop, intrinsically stable strategies to run on this terrain, and not visually guided foot placement, making an important contribution to understanding the context-dependent role of vision in human locomotion.

**\*For correspondence:** m.venkadesan@yale.edu

**Competing interest:** The authors declare that no competing interests exist.

## Introduction

Running on natural terrain is an evolutionarily important human ability (*Carrier et al., 1984*; *Bramble and Lieberman, 2004*), which requires the skillful negotiation of uneven ground (*Lee and Lishman, 1977*; *Warren et al., 1986*). Part of the challenge is planning a path in real-time that navigates around obstacles or sudden steep drops. Even after finding a path around such hazards, the ground would be uneven. Planning the stepping pattern using detailed information of every bump and dip of the ground is typically infeasible on natural trails because the ground is often covered by foliage or grass. But the seemingly slight unevenness, albeit gentler than large obstacles or drops, could have significant consequences to stability. Mathematical modeling predicts that even slightly uneven ground, with peak-to-valley height variations less than the dorso-plantar foot height, could be severely destabilizing unless some form of mitigation strategy is employed to deal with them (*Dhawale et al., 2019*). In this paper, we investigate how human runners deal with these types of undulating uneven ground.

Studies on human walking find that footsteps are visually guided to plan a path through complex, uneven terrain (*Matthis et al., 2018*; *Thomas et al., 2020*; *Bonnen et al., 2021*). Although there are

no similar studies of running on naturalistic uneven terrain, we may expect that vision's role is multi-fold. For example, in the evolutionary context of persistence hunting (*Carrier et al., 1984*; *Bramble and Lieberman, 2004*), vision is needed to track footprints and continuously survey the landscape for prey in addition to dealing with the terrain's unevenness. The potentially competing demands on visual attention—for stability versus other functional goals—is probably more exacting in running than in walking because of the greater speeds involved and the shorter time available to sense and act. Additional important factors to consider on uneven terrain include dynamic stability (*Holmes et al., 2006*; *Dhawale et al., 2019*; *Daley and Biewener, 2006*; *Voloshina and Ferris, 2015*), leg safety (*Birn-Jeffery et al., 2014*), peak force mitigation (*Blum et al., 2014*), and anticipatory leg adjustments (*Birn-Jeffery and Daley, 2012*; *Müller et al., 2015*). However, we presently lack studies of human runners on naturalistic uneven terrain to investigate the role of vision-guided footstep regulation and the subtle regulation of body mechanics for maintaining stability, which motivates the overground running experiments presented in this paper.

In addition to vision, the body's mechanical responses aid stability and are neurally modulated through muscle contractions. These mechanical properties have been studied theoretically, and experimental data have been interpreted, through the lens of models that approximate the runner as a point-like mass on a massless leg, commonly referred to as the spring-legged inverted pendulum (SLIP) model (*Seyfarth et al., 2002*; *Daley et al., 2006*; *Geyer et al., 2006*; *Birn-Jeffery et al., 2014*; *Müller et al., 2016*; *Seethapathi and Srinivasan, 2019*). SLIP models have hypothesized multiple stabilization strategies for terrain with random height variations, several of which have found experimental support: higher leg retraction rates (*Karssen et al., 2015*), wider lateral foot placement (*Voloshina and Ferris, 2015*; *Mahaki et al., 2019*), and the possible use of vision to guide foot placement (*Birn-Jeffery and Daley, 2012*). But SLIP models do not help understand the effect of slope variations because the ground force is constrained to always point to the center of mass irrespective of whether the foot contacts the ground on a level or sloping region. That is a consequence of the zero moment of inertia about the center of mass for SLIP models. Analyses of models with non-zero moment of inertia show that both height and slope variations are detrimental to stability, with slope being more destabilizing (*Dhawale et al., 2019*), reminiscent of common experience among runners.

Understanding why slope variations degrade stability could generate hypotheses and testable predictions for how human runners deal with stability on naturalistic uneven terrain. The mathematical analyses of *Dhawale et al., 2019*, find that random variations in slope lead to step-to-step fluctuations in the fore-aft ground impulse. For steady forward running, the net forward impulse should be zero for every step. But small step-to-step random variation of the fore-aft ground impulse leads to a gradual accumulation of sagittal plane angular momentum, which ultimately destabilizes the runner. However, the rate at which the destabilizing angular momentum builds up depends on where on the terrain the foot lands and how the body responds to landing on the ground, thus suggesting two mitigating strategies. One strategy is to minimize the fore-aft impulse that is experienced at touch down, which has the effect of significantly slowing down the fluctuation-induced build-up of destabilizing angular momentum. This can be achieved by reducing the forward speed of the foot at touchdown via leg retraction and by reducing limb compliance so that the momentum of the rest of the body contributes lesser to the fore-aft impulse. Another strategy is to try and land primarily on local maxima or other flat regions of the terrain so that the destabilizing influence of random slope variations is reduced. The experimental assessment of these two strategies is the topic of this paper.

Most past experimental studies of uneven terrain running have used step-like blocks to show how humans and animals deal with height variations on the ground (*Daley et al., 2006*; *Müller et al., 2015*). Later work modified the terrain design to use blocks that were narrow enough so that the foot had to span more than one fore-aft block, leading the foot to be randomly tilted during foot flat (*Voloshina and Ferris, 2015*). Specifically, the blocks were of three different heights (labeled A, B, and C), which leads to six possible height difference pairings (AB, BA, AC, CA, BC, CB). In natural terrain, the variation in slope is continuously graded, which would allow for more variation in the foot flat angle. Moreover, as hypothesized by theoretical analysis (*Dhawale et al., 2019*), it is not only the foot angle that affects whole body dynamics, but the force direction from the ground also matters. In this regard, the natural terrain may differ from the block design, particularly during initial contact and push-off when only a small region of the foot makes contact with the ground. During that time, the block design would not influence the ground forces like the sloped ground of natural undulating terrain would. Moreover, complex terrain

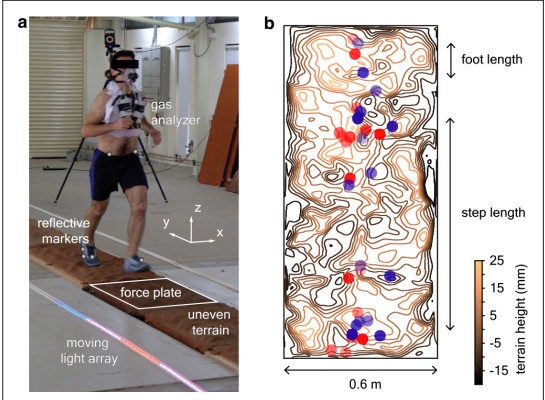

**Figure 1.** Uneven terrain experiments. (**a**) We conducted human-subject experiments on flat and uneven terrain while recording biomechanical and metabolic data. The reflective markers and the outline of the force plate are digitally exaggerated for clarity. (**b**) Footsteps were recorded to determine whether terrain geometry influences stepping location, illustrated here by a mean-subtracted contour plot of terrain height for an approximately 6 foot segment of uneven II overlaid with footsteps (location of the heel marker). Blue and red circles represent opposite directions of travel and transparency level differentiates trials.

The online version of this article includes the following source data for figure 1:

**Source data 1.** Dimensional mass and leg length of every subject.

---

types may be required to capture the range of strategies used to run on naturalistic uneven terrain. This is suggested by studies that examine walking on a variety of outdoor terrain and show that stride variability and energetics significantly depend on terrain complexity (*Kowalsky et al., 2021*). Undulating uneven terrain have been studied in the context of walking (*Kent et al., 2019*; *Kowalsky et al., 2021*), but not running. So there is a need for experiments to study running on undulating terrain with continuously varying slopes to expand the current understanding of how uneven terrain affects stability. In this paper we experimentally assess foot placement patterns, fore-aft ground impulses, stepping kinematics, and metabolic power consumption on undulating uneven terrain whose unevenness is akin to running trails (*Figure 1*).

## Methods
### Protocol and experimental measurements

We conducted overground running experiments with nine subjects (eight men, one woman; age 23–45 years, body mass $66.1 \pm 8.5$ kg, leg length $0.89 \pm 0.04$ m, reported as mean ± SD). All subjects were able-bodied, ran approximately 30 km per week, and had run at least one half-marathon or marathon within the previous year. Experiments were conducted at the National Centre for Biological Sciences, Bangalore, India, with informed consent from the volunteers, and IRB approval (TFR:NCB:15_IBSC/2012).

Subjects ran back-and-forth on three 24 m long and 0.6 m wide tracks (*Figure 2a*). In addition to a *flat* track, we used two custom-made uneven tracks, *uneven I* and *uneven II*, which had increasing unevenness. Uneven I and uneven II had peak-to-valley height differences (amplitude) of $18 \pm 6$ and $28 \pm 11$ mm (mean ± SD), respectively, and peak-to-peak horizontal separation (wavelength) of $102 \pm 45$ and $108 \pm 52$ mm, respectively (*Figure 2b, c and d*). We recorded kinematics using an 8-camera motion capture system (Vicon Inc., Oxford, UK) at 300 frames per second and measured the ground reaction forces at 600 Hz using two force plates (AMTI Inc., model BP600900) embedded beneath the center of the track. The cameras recorded an approximately 10 m long segment of the center of the track. Breath-by-breath respirometry was also recorded by a mobile gas analyzer (Oxycon Mobile, CareFusion Inc.).

A single trial consisted of a 3 min period of standing when the resting metabolic rate was recorded followed by subjects running back-and-forth on the track for at least 8 min and up to 10 min, dictated by $VO_2$ reading equilibration time and the subject's ability to maintain speed over the course of the

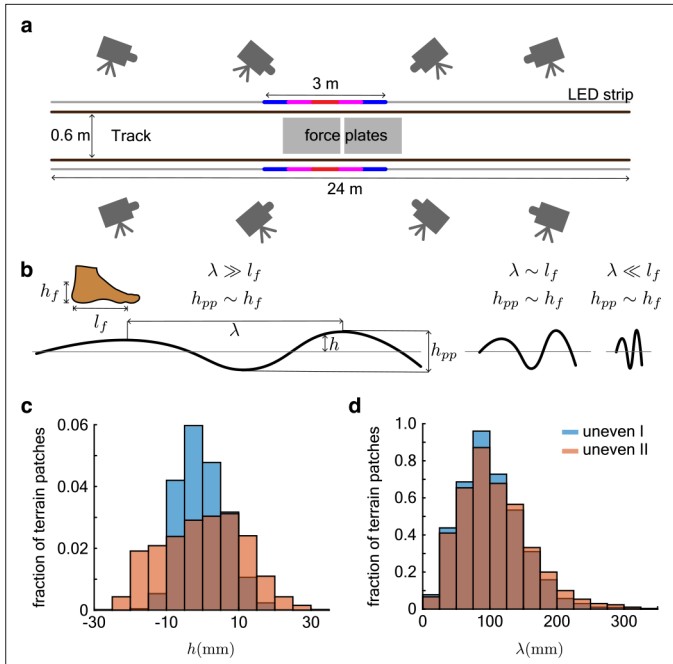

**Figure 2.** Details of the experiment design. (**a**) Schematic of the running track, camera placement, force plate positions and the LED strip with a 3 m illuminated section. (**b**) The terrain was designed so that the range of its height distribution $h$ was comparable to ankle height $h_{pp} \sim h_f$ and peak-to-peak distances $\lambda$ (along the length of the track) were comparable to foot length $\lambda \sim l_f$. (**c**) Histograms of the mean subtracted heights $h$ of the uneven terrain. (**d**) Histograms of the peak-to-peak separation $\lambda$ of the uneven terrain.

trial. Each subject ran on all three terrains, with the order randomized. We controlled the running speed using a moving light array in 24 m long LED strips laid on either side of the track (*Figure 2a*). Subjects were instructed to stay within the bounds of a 3 m illuminated segment of the LED strip that traveled at 3 m/s. This speed was chosen as it was comfortable for all subjects and lies within the endurance running speed range for humans (*Bramble and Lieberman, 2004*). Running speed fluctuated within a trial, however mean speed as well as speed variability were consistent across terrain types (see Results for details). Subjects were provided with standardized, commercially available running shoes.

## Uneven terrain

Terrain unevenness was heuristically specified so that peak-to-valley height variations were approximately equal to the height of the malleolus while standing barefoot on level ground, and peak-to-peak horizontal distances were similar to foot length (*Figure 2b*). Large terrain height variations may elicit obstacle avoidance strategies, which is not the subject of this paper, and peak-to-peak horizontal separation longer than the step length may make the slope variation too gentle. Conversely, small height variations that are similar to the heel pad thickness, and peak-to-peak horizontal separation that is smaller than the foot length, will likely be smoothed out by foot and sole compliance (*Venkadesan et al., 2017*).

The uneven terrains were constructed by Mars Adventures Inc. (Bangalore, India) by laying fiber glass over heuristically created contours. Epoxy was used to harden the fiber glass sheets into a stiff shell which was coated with a slurry of sand and epoxy to create a surface that texturally resembles weathered rock. The width at the ends of the uneven track were broadened to approximately 1 m to allow for runners to change direction while remaining on the terrain. The terrain was then digitized using a dense arrangement of reflective markers that were recorded by the motion capture system.

## Kinematics

Foot kinematics were recorded using fiducial markers that were fixed to the shoes over the calcaneus, second distal metatarsal head, and below the lateral malleolus. Markers were attached to the hip, over

the left and right lateral superior iliac spine, and the left and right posterior superior iliac spine. The mean position of the hip markers was used to estimate the center of mass location.

Stance is defined as when the heel marker's forward velocity was minimized and its height was within 15 mm of the marker's height during standing. The threshold of 15 mm was chosen to account for terrain height variations so that stance may be detected even when the heel lands on a local peak of the uneven terrain.

The center of mass forward speed $v = d_{step}/t_{step}$ is found from the distance $d_{step}$ covered by the center of mass in the time duration $t_{step}$ between consecutive touchdown events. Leg angle at touchdown is defined as the angle between the vertical and the line formed by joining the heel marker to the center of mass. Virtual leg length at touchdown is defined as the distance between the heel marker and the center of mass. Foot length $l_f$ is defined as the average distance in the horizontal plane between the toe and heel marker, across all subjects. The center of mass trajectory during stance was fitted with a regression line in the horizontal plane. The step width is twice the distance of nearest approach of the stance foot from the regression line. This definition allows for the runner's center of mass trajectory to deviate while preserving a definition of step width that is consistent with those previously used (**Donelan et al., 2001**; **Arellano and Kram, 2011**). We estimated meander, i.e., the deviation of the center of mass from a straight trajectory, using $(d - d_0)/d_0$, where $d$ is the distance covered by the center of mass in the horizontal plane during a single run across the length of the track and $d_0$ is the length of the straight-line fit to the center of mass trajectory. Foot velocity or center of mass velocity at landing were calculated by fitting a cubic polynomial to the heel marker trajectory or center of mass trajectory, respectively, in a 100 ms window before touchdown, and calculating the time derivative of the fitted polynomial at the moment just prior to touchdown. Leg retraction rate $\omega$ is determined using $\omega = v_f/\|\mathbf{l}\|$, where $v_f$ is the component of the foot's relative velocity with respect to the center of mass that is perpendicular to the virtual leg vector $\mathbf{l}$ (vector joining heel to center of mass).

Step width, step length, and virtual leg length at touchdown are normalized by the subject's leg length, defined as the distance between the greater trochanter and lateral malleolus.

To correct for slight angular misalignments between the motion capture reference frame and the long axis of the running track, we align the average CoM trajectory over the entire track length to be parallel to the y-axis of the motion capture reference frame. This correction reflects the experimental observation that the subjects run along the center of the track.

## Kinetics

Force plate data were low-pass filtered using an eighth order, zero-phase, Butterworth filter with a cut-off frequency of 270 Hz. Touchdown on the force plates was defined by a threshold for the vertical force of four standard deviations above the mean unloaded baseline reading.

The forward collision impulse, defined as the maximal decelerating fore-aft impulse $J_y^*$, was found by integrating the fore-aft component $F_y$ of the ground reaction force during the deceleration phase as

$$J_y^* = \max_t \left| \int_0^t F_y(\tau)\, d\tau \right|. \tag{1}$$

We normalized $J_y^*$ by the aerial phase forward momentum $mv_y$, where $v_y$ is the forward speed of the center of mass during the aerial phase.

## Energetics

Net metabolic rate is defined as the resting metabolic power consumption subtracted from the power consumption during running and normalized by the runner's mass. Metabolic power consumption is determined using measurements of the rate of $O_2$ consumption and $CO_2$ consumption using formulae from **Brockway, 1987**. For running, this is calculated after discarding the first 3 min of the run to eliminate the effect of transients. The resting metabolic power consumption is calculated after discarding the first minute of the standing period of the trial. Data from each trial were visually inspected to ensure that the rates of $O_2$ consumption and $CO_2$ production had reached a steady state, seen as a plateau in the data trace.

## Shuttle running

Of the total track length of 24 m, a 1.2 m turnaround segment was designed at each end to facilitate the subjects to reverse their running direction without stepping off the track. These end segments were 1 m wide, which was broader than the rest of the track that was only 0.6 m wide. The runners would reach the end of the track and turn around promptly. Guiding light bars that controlled the running speed would be half 'absorbed' into the end before reversing direction, which allowed for sufficient time for the subjects to turn around while still maintaining the same average speed. The subjects were given, and took, around 0.5 s to turn around. The subjects ran at a steady speed within the capture volume that covers the middle 10 m of the track (see Results for details). The cameras could not capture the ends of the track but the experimenters observed that the subjects stayed within the moving light bar through the 21.6 m long straight portion of the track. The experimental protocol used in this study was tuned through pilot trials involving the authors of this manuscript and two initial subjects. The data from these pilot trial subjects are not part of the reported results in this manuscript.

## Foot stepping analysis

### Directed foot placement scheme

The runners' foot landing locations were compared to a Markov chain Monte Carlo (MCMC) model which finds stepping locations with the lowest terrain unevenness subject to constraints of matching experimentally measured stepping kinematics. All participants were heel-strike runners on all terrain types, as judged from the double peak in the vertical ground reaction force profile. Therefore, the stepping model sampled the terrain in rear-foot sized patches, which we define to be 95 mm × 95 mm (dimensions are chosen to be half the size of the foot length, 190 mm). The interquartile range of heights ($h_{\mathrm{IQR}}$) in each patch was used as a measure of its unevenness.

Starting from an initial position $(x_i, y_i)$, the model takes the next step to $(x_{i+1}, y_{i+1})$ in the following stages: *open-loop stage*, *minimization stage*, and a *noise process* given by,

open-loop stage:

$$\hat{x}_{i+1} = x_i + (-1)^i s_w, \ \hat{y}_{i+1} = y_i + (-1)^j s_l. \tag{2}$$

Minimization stage:

$$\begin{aligned}
(x'_{i+1}, y'_{i+1}) &= \arg\min_{(x,y)} t(x, y), \\
x &\in [\hat{x}_{i+1} - \sigma_{sw}, \hat{x}_{i+1} + \sigma_{sw}], \\
y &\in [\hat{y}_{i+1} - \sigma_{sl}, \hat{y}_{i+1} + \sigma_{sl}].
\end{aligned} \tag{3}$$

Noise process:

$$\begin{aligned}
x_{i+1} &= x'_{i+1} + \eta_x, \ y_{i+1} = y'_{i+1} + \eta_y, \\
&\text{where } \eta_x \sim v_M(1, 0, \sigma_{sw}), \ \eta_y \sim v_M(1, 0, \sigma_{sl}).
\end{aligned} \tag{4}$$

In the open-loop stage, the model takes a step forward and sideways dictated by the experimentally measured mean step length $s_l$ and mean step width $s_w$, respectively. The exponent $j$ is either 0 or 1 and keeps track of the direction of travel. The function $t(x, y)$ evaluates the interquartile range of heights of a rear-foot sized patch centered around position $(x, y)$. In the minimization step, the model conducts a bounded search about $(\hat{x}_{i+1}, \hat{y}_{i+1})$ for the location that minimizes $t(x, y)$. The search region is defined by the standard deviations of the measured step width $\sigma_{sw}$ and step length $\sigma_{sl}$. To perform the minimization, a moving rear-foot sized window with step-sizes of $\sigma_{sw}/10$ along the width of the track and $\sigma_{sl}/10$ along its length are used to evaluate $t(x, y)$ at various candidate stepping locations within the search region. The step-sizes for translating the moving window were chosen because they were much smaller than typical terrain features and thus the landing location with the lowest unevenness $(x'_{i+1}, y'_{i+1})$ was determined by the terrain properties, not model parameters. To simulate sensorimotor noise, the location of this minimum $(x'_{i+1}, y'_{i+1})$ is perturbed by random variables $\eta_x, \eta_y$. The random variables are drawn from von Mises distributions with $\kappa = 1$, centered about zero, and scaled so that the base of support for the distributions are $\sigma_{sw}$ and $\sigma_{sl}$, respectively.

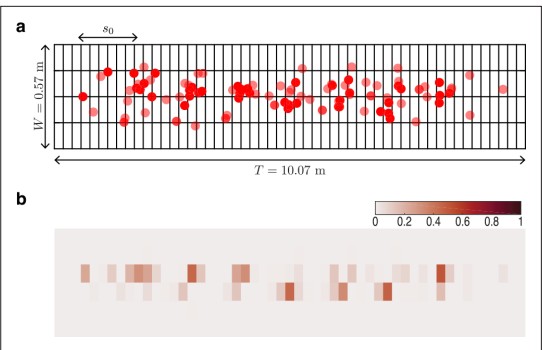

**Figure 3.** Foot placement analysis. (**a**) Red circles denote footstep locations (392 footsteps) in the '$x - y$' plane for a representative trial on uneven II. The grid spacing is 190 mm along the length of the track and 95 mm along its width. Step length $s_0$ is shown for reference. $T$ is the length of the capture volume and $W$ is the width of the track. Note that the $x$ and $y$ axes of this figure are not to the same scale. (**b**) The probability of landing on a foot-sized region of the track is quantified by the foot placement index *Equation 5* shown as a heatmap with the color bar at the top left.

The online version of this article includes the following source data and figure supplement(s) for figure 3:

**Source data 1.** Footstep counts for each subject on all terrain.

**Figure supplement 1.** Subject-wise foot placement patterns.

At the ends of the track, the $x$ position of the runner is reset so that the runner is at the center of the track, and the direction of travel is reversed ($j$ value is toggled). We simulate for 100,000 steps to ensure that reported terrain statistics at footstep locations as well as step length and step width converge, i.e., errors between simulations in these parameters are less than 1% of their mean value.

## Quantifying foot placement patterns

We used a second analysis of footstep patterns that correlated the foot landing probability with terrain unevenness. To perform this analysis, we define a foot placement index to estimate the probability that the runner's foot lands within a foot-sized patch of the track. To calculate this index, we first divide the terrain into a grid of 0.5 foot lengths × 1.0 foot lengths cells, with the longer side of the cell along the length of the track (*Figure 3a*). We count the number of footsteps $f_{i,j}$ in each cell $c_{i,j}$, where $i$ indexes the position of the cell along the length of the track and $j$ indexes the position of cell transverse to the track. The point of landing is determined by the location of the heel marker. Even if the fore-foot crosses over the adjacent cell boundary, the location of the heel marker uniquely specifies the landing cell identity. We also define step length-sized neighborhoods that contain cell $c_{i,j}$ which are one step-length long and as wide as the track. Each such neighborhood has a cumulative footstep count $S_i$ that depends on the longitudinal location $i$ of the cell. The average across all such step length-sized neighborhoods that contain cell $c_{i,j}$ is $S$. This average $S$ is used to normalize each $f_{i,j}$ to yield the foot placement index $p_{i,j}$ according to,

$$p_{i,j} = \frac{f_{i,j}}{S}. \tag{5}$$

The index $p_{i,j}$ measures the fraction of times a foot lands in cell $c_{i,j}$ compared to all other cells that are within a step length distance of it (*Figure 3b*). If runners were perfectly periodic with no variation in footstep location from one run over the terrain to the next, $p_{i,j} = 1$ for cells on which subjects stepped and $p_{i,j} = 0$ otherwise. If, however, stepping location was the result of a uniform random process, $p_{i,j}$ would be a constant for every cell of the terrain and equal to the reciprocal of the number of cells in a step-length sized box. Heat maps of the foot placement index $p_{i,j}$ are shown in *Figure 3—figure supplement 1*. We report the total number of footsteps recorded for each trial in *Figure 3—source data 1*.

To probe foot placement strategies we determine whether the foot placement index $p_{i,j}$ correlates with the median height or the interquartile range of heights within the cell $c_{i,j}$. Positive correlation with the median height would indicate stepping on local maxima that are flatter than the surrounding, and

negative correlation with the interquartile range would indicate stepping on flatter regions with more uniform height. We test this hypothesis through the use of a statistical model described in 'Statistical analysis and reporting'.

## Collision model

To delineate the relative contributions of joint stiffness and forward foot speed to the fore-aft impulse, we model the impulse due to the foot-ground interaction. In the model, a planar three-link chain represents the foot, shank, and thigh, and a fourth link represents the torso. Following *Dempster, 1955*, all masses and lengths are expressed as fractions of the body mass and leg length, respectively. This model builds upon the leg collision model of *Lieberman et al., 2010*, by including additional segments representing the thigh and torso and calculating the fore-aft collisional impulse. The collision is assumed to be instantaneous and inelastic, with a point-contact between the leg and the ground. Such collision models are widely used to capture the stance impulse due to ground forces in walking (*Donelan et al., 2002*; *Ruina et al., 2005*) and running (*Srinivasan and Ruina, 2006*; *Dhawale et al., 2019*). Because the collision is assumed to be instantaneous, only infinite forces contribute to the impulse (*Chatterjee and Ruina, 1998*; *Lieberman et al., 2010*). Therefore, to investigate the effect of joint compliance, we model the hinge joints connecting the links as either infinitely compliant or perfectly rigid. The advantage of these contact models is their ability to accurately capture the impulse without the numerous additional parameters needed to represent the complete force-time history when contact occurs between two bodies (*Chatterjee and Ruina, 1998*).

We use experimental data on center of mass velocity and leg retraction rate just prior to landing, along with the leg angle at touchdown, to compute a predicted collisional impulse. Because all our runner's were heel-strikers, we use foot-strike index $s = 0.15$ for the collision calculations (*Lieberman et al., 2010*). The foot-strike index ranges from 0 for heel strikes to 1 for forefoot strikes and encodes the runner's foot strike pattern. The ratio of the collisional impulse to the measured whole body momentum just prior to landing is calculated for the model at the two joint stiffness extremes and compared with experimental measurements of the normalized fore-aft impulse. By analyzing the collisional impulse for these two extremes of joint stiffness, we isolate the contributions to the fore-aft impulse arising from varying the joint stiffness versus varying the forward foot speed at landing.

### Notation

Notation used in this section is as follows. Scalars are denoted by italic symbols (e.g. $I$ for the moment of inertia), vectors by bold, italic symbols ($\mathbf{v}$ for velocity), and points or landmarks in capitalized non-italic symbols (such as center of mass G in *Figure 4a*). Vectors associated with a point, such as the velocity of center of mass G are written as $\mathbf{v}_G$, with the upper-case alphabet in the subscript specifying

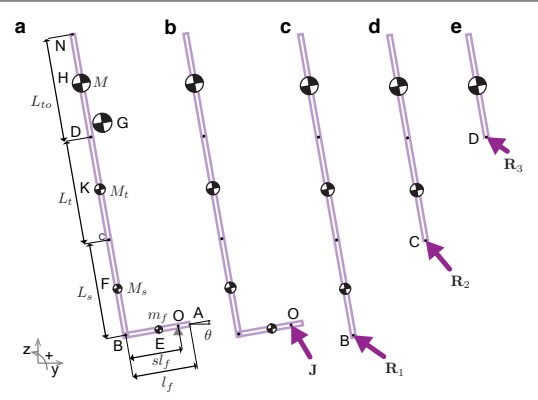

**Figure 4.** Model for estimating fore-aft collision impulses from kinematic data. (**a**) A four-link model of the foot (**A–B**), shank (**B–C**), thigh (**C–D**), and torso (**D–N**) moving with center of mass velocity $\mathbf{v}_G^-$ and angular velocity $\mathbf{\Omega}^-$ collides with the ground at angle $\theta$. (**G**) represents the center of mass. Leg length and body mass are obtained from data and scaled according to *Dempster, 1955*, to obtain segment lengths and masses. Free-body diagrams show all non-zero external impulses: (**b**) collisional impulse $\mathbf{J}$ acting at O, and panels (**c, d, e**) show reaction impulses $\mathbf{R}_1$, $\mathbf{R}_2$, and $\mathbf{R}_3$ acting at B, C, and D, respectively.

the point in the plane. Moment of inertia variables are subscripted with '/A' representing the moment of inertia computed about point A, such as $I_{/G}$ representing the moment of inertia about the center of mass G. Position vectors are denoted by $\mathbf{r}_{A/B}$ which denotes the position of point A with respect to point B. Variables just before the collision with the terrain are denoted by the superscript '−', and just after the collision by the superscript '+'. Equations with variables that have no superscript apply throughout stance.

## Rigid joints

Consider the L-shaped bar (*Figure 4a*) falling with velocity $\mathbf{v}_G^- = v_y^- \hat{j} + v_z^- \hat{k}$ and angular velocity $\mathbf{\Omega}^- = \omega^- \hat{i}$. Upon contact with the ground, the point O on the foot instantly comes to rest and the center of mass translational and angular velocities change to $\mathbf{v}_G^+ = v_y^+ \hat{j} + v_z^+ \hat{k}$, $\mathbf{\Omega}^+ = \omega^+ \hat{i}$. Due to the instantaneous collision assumption, finite forces like the gravitational force do not contribute to the collisional impulse, and the ground reaction force at point O leads to the impulse $\mathbf{J}$ (*Figure 4b*). Angular momentum balance about the contact point O yields the relationship between pre and post collision velocities,

$$M_b \mathbf{r}_{G/O} \times \mathbf{v}_G^- + I_{/G} \mathbf{\Omega}^- = M_b \mathbf{r}_{G/O} \times \mathbf{v}_G^+ + I_{/G} \mathbf{\Omega}^+, \tag{6a}$$

$$\mathbf{v}_G = \mathbf{v}_O + \mathbf{\Omega} \times \mathbf{r}_{G/O}, \tag{6b}$$

$$\text{where } \mathbf{v}_O^+ = 0. \tag{6c}$$

The total mass $M_b$ is the sum of the masses of the torso $M$, thigh $M_t$, shank $M_s$, and foot $m_f$. We solve for $\omega^+$ in *Equation 6a* and obtain the post-collision center of mass velocity $\mathbf{v}_G^+$ using *Equation 6b*. From this, the collision impulse $\mathbf{J}$ and the normalized fore-aft collisional impulse $|J_y^*|/J_b$ are calculated using,

$$\mathbf{J} = M_b(\mathbf{v}_G^+ - \mathbf{v}_G^-), \tag{7a}$$

$$J_y^* = \mathbf{J} \cdot \hat{j}, \tag{7b}$$

$$\text{and } J_b = M_b(\mathbf{v}_G^- \cdot \hat{j}). \tag{7c}$$

## Compliant joints

If the L-bar has compliant joints, then the post-collision velocities for each segment may vary. Therefore, we write additional angular momentum balance equations for each segment to solve for the post-collision state. Since the only non-zero external impulse acting on the shank, thigh, and torso segments is the reaction impulse $\mathbf{R}_1$ acting at B (*Figure 4c*), the only non-zero external impulse on the thigh and torso portion of the leg is the reaction impulse $\mathbf{R}_2$ acting at C (*Figure 4d*), and the only non-zero external impulse acting on the torso portion of the leg is the reaction impulse $\mathbf{R}_3$ acting at D (*Figure 4e*), we write angular momentum balance equations for the entire body and these three segments as

$$
\begin{aligned}
M_b \mathbf{r}_{G/O} \times v_G^- + I_{/G} \mathbf{\Omega}^- = {} & m_f \mathbf{r}_{E/O} \times \mathbf{v}_E^+ + I_{/E} \mathbf{\Omega}_E^+ + \\
& M_s \mathbf{r}_{F/O} \times \mathbf{v}_F^+ + I_{/F} \mathbf{\Omega}_F^+ + \\
& M_t \mathbf{r}_{K/O} \times \mathbf{v}_K^+ + I_{/K} \mathbf{\Omega}_K^+ + \\
& M \mathbf{r}_{H/O} \times \mathbf{v}_H^+ + I_{/H} \mathbf{\Omega}_H^+,
\end{aligned} \tag{8a}
$$

$$
\begin{aligned}
M_s \mathbf{r}_{F/B} \times \mathbf{v}_F^- + M_t \mathbf{r}_{K/B} \times \mathbf{v}_K^- + {} & \\
M \mathbf{r}_{H/B} \times \mathbf{v}_H^- + (I_{/F} + I_{/K} + I_{/H}) \mathbf{\Omega}^- = {} & M_s \mathbf{r}_{F/B} \times \mathbf{v}_F^+ + I_{/F} \mathbf{\Omega}_F^+ + \\
& M_t \mathbf{r}_{K/B} \times \mathbf{v}_K^+ + I_{/K} \mathbf{\Omega}_K^+ + \\
& M \mathbf{r}_{H/B} \times \mathbf{v}_H^+ + I_{/H} \mathbf{\Omega}_H^+,
\end{aligned} \tag{8b}
$$

$$
\begin{aligned}
M_t \mathbf{r}_{K/C} \times v_K^- + M \mathbf{r}_{H/C} \times \mathbf{v}_H^- + {} & \\
(I_{/K} + I_{/H}) \mathbf{\Omega}^- = {} & M_t \mathbf{r}_{K/C} \times v_K^+ + I_{/K} \mathbf{\Omega}_K^+ + \\
& M \mathbf{r}_{H/C} \times \mathbf{v}_H^+ + I_{/H} \mathbf{\Omega}_H^+
\end{aligned} \tag{8c}
$$

$$M \mathbf{r}_{H/D} \times \mathbf{v}_H^- + I_{/H} \mathbf{\Omega}^- = M \mathbf{r}_{H/D} \times \mathbf{v}_H^+ + I_{/H} \mathbf{\Omega}_H^+ \tag{8d}$$

where $I_{/E}, I_{/F}, I_{/K}, I_{/H}$ are moments of inertia of the foot, shank, thigh, and torso segments, respectively, about their centers. The linear and angular velocities of the foot ($\mathbf{v}_E, \mathbf{\Omega}_E$), shank ($\mathbf{v}_F, \mathbf{\Omega}_F$), thigh ($\mathbf{v}_K, \mathbf{\Omega}_K$), and torso ($\mathbf{v}_H, \mathbf{\Omega}_H$) are related to the velocity of the contact point O as

$$\mathbf{v}_E = \mathbf{v}_O + \mathbf{\Omega}_E \times \mathbf{r}_{E/O}, \tag{9a}$$

$$\mathbf{v}_F = \mathbf{v}_O + \mathbf{\Omega}_E \times \mathbf{r}_{B/O} + \mathbf{\Omega}_F \times \mathbf{r}_{F/B}, \tag{9b}$$

$$\mathbf{v}_K = \mathbf{v}_O + \mathbf{\Omega}_E \times \mathbf{r}_{B/O} + \mathbf{\Omega}_F \times \mathbf{r}_{C/B} + \mathbf{\Omega}_K \times \mathbf{r}_{K/C}, \tag{9c}$$

$$\mathbf{v}_H = \mathbf{v}_O + \mathbf{\Omega}_E \times \mathbf{r}_{B/O} + \mathbf{\Omega}_F \times \mathbf{r}_{C/B} + \mathbf{\Omega}_K \times \mathbf{r}_{D/C} + \mathbf{\Omega}_H \times \mathbf{r}_{H/D}, \tag{9d}$$

$$\text{where } \mathbf{v}_O^- = \mathbf{v}_G^- + \mathbf{\Omega}^- \times \mathbf{r}_{O/G}, \tag{9e}$$

$$\text{and } \mathbf{v}_O^+ = 0. \tag{9f}$$

Simultaneously solving *Equations 8a*, *b*, *c*, *d*–*9a*, *b*, *c*, *d*, *e*, *f* yields the post-collision velocities for each segment of the L-bar. From these, we calculate the normalized fore-aft collision impulse for the compliant model using *Equation 7a*, *b*, *c*.

## Statistical methods

### Sample size

Sample size could refer to the number of subjects or the number of footsteps that were used in the analyses. The number of subjects recruited was informed by typical participant numbers that were used in similar past studies (*Donelan et al., 2004*; *Voloshina and Ferris, 2015*; *Seethapathi and Srinivasan, 2019*). There is an additional consideration for sufficiency of sample numbers for the foot placement analysis. The steps should densely sample the approximately 10 m long central region of the track, where the motion capture system was recording from. The 5262 recorded steps (2526 on uneven I, 2736 on uneven II) are sufficient to densely sample the measurement region assuming a rear-foot sized patch for each step.

### Statistical analysis and reporting

Measures of central tendency (mean or median) and variability (standard deviation or interquartile range) of the distributions of step width, step length, center of mass speed, forward foot speed at landing, fore-aft impulse, virtual leg length at touchdown, leg angle at touchdown, net metabolic rate, and meander are reported for each trial.

We use three different linear mixed models to determine (a) whether gait variables vary with terrain type, (b) whether leg angle at touchdown and decelerating fore-aft impulses covary with forward foot speed at touchdown, and (c) whether the foot placement index $p_{i,j}$ (*Equation 5*) correlates with the median height or the interquartile range of heights within the terrain region at landing. The statistical models are run using the lmerTest package in R (*Kuznetsova et al., 2017*). We use a linear mixed-model fit by restricted maximum likelihood t-tests with Satterthwaite approximations to degrees of freedom. An ANOVA on the first model tests for the effect of the terrain factor, an ANCOVA on the second model tests for the effect of the terrain factor and the covariate forward foot speed, and an ANCOVA on the third model tests whether the probability of landing on a terrain patch $p_{i,j}$ significantly covaries with the height or unevenness of that terrain patch. Post-hoc pairwise comparisons, where relevant, are performed using the emmeans package in RStudio with p-values adjusted according to Tukey's method.

A measure of central tendency or variability within a trial is the dependent variable $y$ for the first linear mixed model. There are 27 observations for the dependent variable $y$ corresponding to each trial (nine subjects running on three terrain). Terrain is the fixed factor and subjects are random factors in the model given by

$$y_{ij} = (\beta_0 + \mu_j) + \beta_i \text{terrain}_i + \epsilon_{ij}, \tag{10}$$

where $i = 1, 2$ and $j = 1 \ldots 9$. The intercept $\beta_0$ (value of $y$ on flat terrain) and parameters $\beta_i$ for uneven I and uneven II are estimated for this model. The random factor variables $\mu_j$ are assumed to be normally distributed about zero and account for inter-subject variability of the intercept. The model residuals are $\epsilon_{ij}$ which are also assumed to be normally distributed about zero.

The second linear mixed model uses stepwise data where each step is grouped by subject and terrain type. Each of the 1086 steps in this dataset contains a value for subject number, terrain type, touchdown leg angle, decelerating fore-aft impulse, and forward foot speed at touchdown. The linear model for the dependent variable $y$ (touchdown leg angle or fore-aft impulse) is

$$y_{ij} = (\beta_0 + \mu_{1j}) + \beta_i \text{terrain}_i + (\beta_f + \mu_{2j} + \nu_i)\text{footspeed} + \epsilon_{ij} \tag{11}$$

where $i = 1, 2$ and $j = 1 \ldots 9$. Like in *Equation 10*, the model estimates the intercept $\beta_0$, i.e., the value of $y$ on flat terrain when foot speed = 0, $\beta_i$ for terrain factor, and the slope $\beta_f$ for the dependence of $y$ on forward foot speed at touchdown. The variable $\mu_{1j}$ account for inter-subject variability of the intercept, and the variables $\mu_{2j}$ and $\nu_i$ account for inter-subject and terrain-specific variability of the slope $\beta_f$, respectively. The residuals $\epsilon_{ij}$ are assumed to be normally distributed.

Using a dataset of 5262 steps from all subjects on uneven I and uneven II, we extract 1515 landing probabilities (as detailed in 'Quantifying foot placement patterns'). To test whether runners aimed for terrain regions with low unevenness, we use a linear mixed model of the form,

$$y_{kl} = (\mu_{1l} + \nu_{1k}) + (\mu_{2l} + \nu_{1k})\text{terr} + \epsilon_{kl} \tag{12}$$

where $k = 1, 2$ for the two uneven terrain and $l = 1 \to 9$ for the nine subjects. The dependent variable $y$ is the probability of landing in a foot-sized cell $p_{i,j}$ and the independent variable 'terr' refers to the median terrain height of the cell or the interquartile range of heights within the cell. The variables $\mu_{1l}$ accounts for subject-specific variability in the terrain-specific intercept $\nu_{1k}$. The variables $\mu_{2l}$ accounts for subject-specific variability in the terrain-specific slope $\nu_{2k}$.

## Nondimensionalization

Following *Alexander and Jayes, 1983*, we express lengths in units of leg length $\ell$ and speed in units of $\sqrt{g\ell}$, where $g$ is acceleration due to gravity. Statistically significant post-hoc comparisons are additionally reported in dimensional units using $g = 9.81 \text{ m/s}^2$, and the mean of the measurements across subjects, namely, $\ell = 0.89$ m and $m = 66.1$ kg.

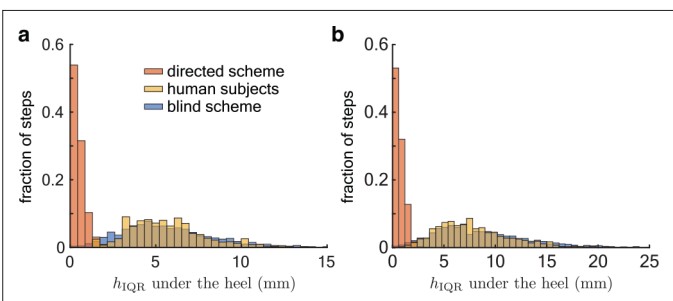

**Figure 5.** Foot placement on uneven terrain. Histogram of the interquartile range of heights ($h_{IQR}$) at footstep locations for the directed sampling scheme (red), experiments (yellow), and the blind sampling scheme (blue) on (a) uneven I (2526 footsteps) and (b) uneven II (2736 footsteps). Note that $h_{IQR}$ varies over a greater range on uneven II.

The online version of this article includes the following source data and figure supplement(s) for figure 5:

**Source data 1.** Output of the Markov chain sampling (directed scheme) of the Uneven I terrain.

**Source data 2.** Output of the Markov chain sampling (directed scheme) of the Uneven II terrain.

**Source data 3.** Output of the uniform random sampling (blind scheme) of the Uneven I terrain.

**Source data 4.** Output of the uniform random sampling (blind scheme) of the Uneven II terrain.

**Source data 5.** Subject-wise, per-step data of the terrain height at foot landing locations on the Uneven I terrain.

**Source data 6.** Subject-wise, per-step data of the terrain height at foot landing locations on the Uneven II terrain.

**Figure supplement 1.** Subject-wise foot placement analysis on uneven I.

**Figure supplement 2.** Subject-wise foot placement analysis on uneven II.

**Figure supplement 3.** Subject-wise foot placement analysis.

## Results

### Foot placement on uneven terrain

To test whether real runners prefer to land on flatter patches, the measured footsteps were compared against two extreme models, a null hypothesis of a *blind* runner and an alternative hypothesis of a *directed* runner whose footsteps are selectively aimed at level parts of the terrain. The blind scheme uses a uniform random sample of rear-foot sized patches of the terrain to obtain statistics of the terrain at landing locations. The directed scheme preferentially samples more level patches using an MCMC model ('Directed foot placement scheme' in Methods).

The experimentally measured stepping patterns are the same as the blind scheme on both uneven I and II in terms of the terrain unevenness as quantified by $h_{IQR}$ (human subjects versus blind scheme in *Figure 5*). However, the directed scheme finds substantially more level landing patches, showing that it was possible for the runners to land on more level ground (directed scheme in *Figure 5*). These trends are also borne out in a subject-wise analysis (*Figure 5—figure supplements 1 and 2*).

The directed scheme found more level patches and exhibited decreased variability in step length and step width compared with the experimental data. The mean step length and width of the directed scheme are the same as the experimental data on both uneven I and uneven II. However, the standard deviation of step length decreased by 80% on both uneven I and uneven II compared to experimental measurements. This corresponds to a change of 0.013 and 0.011 m for the mean subject on uneven I and uneven II, respectively. The standard deviation of step width for the directed scheme decreased by 80% (0.0006 m) on uneven I and by 84% (0.005 m) on uneven II compared to experimental measurements.

The overall statistics of the terrain location at foot landing may obscure step-to-step dependence of the foot landing on terrain features. A second analysis of correlating foot landing probability $p_{i,j}$

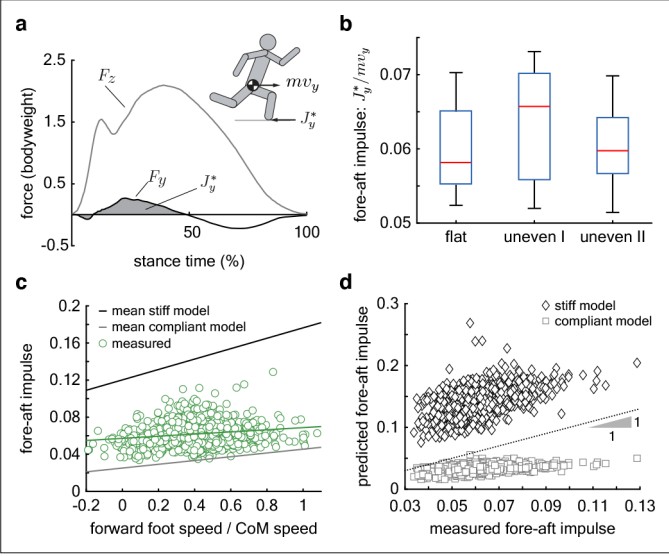

**Figure 6.** Regulation of fore-aft impulses. (**a**) The fore-aft impulse $J_y^*$ (gray shaded area) is found by integrating the measured fore-aft ground reaction force $F_y$ (black curve) during the deceleration phase. (**b**) Mean $\frac{J_y^*}{mv_y}$ for 9 subjects on 3 terrain types. Central red lines denote the median, boxes represent the interquartile range, whiskers extend to 1.5 times the quartile range, and open circles denote outliers. (**c**) Measured $\frac{J_y^*}{mv_y}$ (green circles) versus relative forward foot speed at landing (forward foot speed/center of mass speed) for each step recorded on all terrain types (total 1081 steps). The green line is the regression fit for the data. The dark and light gray lines are the predicted fore-aft impulse for the mean stiff and compliant jointed models, respectively. Per step model predictions in *Figure 6—figure supplement 1*. (**d**) Measured versus predicted fore-aft impulses for every step. The dotted line represents perfect prediction.

The online version of this article includes the following source data and figure supplement(s) for figure 6:

**Source data 1.** Subject-wise, per-step data of fore-aft impulse, foot speed, and touchdown angle.

*Figure 6 continued on next page*

*Figure 6 continued*

**Source data 2.** Per-step data of the measured and predicted fore-aft impulse for the compliant and stiff-leg collision models.

**Figure supplement 1.** Detailed results of the collision analysis.

with the interquartile range of the terrain heights in the foot-sized cell was consistent with results described above and showed no significance (*Table 1*). Taken together, these results indicate that the runners did not guide their footsteps toward flatter areas of the terrain.

## Fore-aft impulses

The fore-aft ground reaction force in stance initially decelerates the center of mass before accelerating it forward (*Figure 6a*). We find that less than $6 \pm 1\%$ (mean ± SD) of the forward momentum is lost during the deceleration phase of stance and there is no dependence on terrain or subject (*Figure 6b*). The low variability of the fore-aft impulse, just 1% of the forward momentum, suggests that it is tightly regulated across runners, terrain, and steps.

The regulation of foot speed is unlikely to be the primary determinant of the low variability in the collision impulse. This is because the dimensionless forward foot speed at touchdown across all terrain varied by nearly 50% of its mean ($0.4 \pm 0.2$, *Table 2*), whereas fore-aft collision impulses varied only by 17% of its mean. A statistical analysis lends further support and shows that the dimensionless fore-aft impulse depends significantly, but only weakly, on the dimensionless forward foot speed at landing (*Table 3*, $p = 0.001$, slope = $0.01 \pm 0.003$).

To further investigate this weak dependence of the retarding impulse on foot speed, we analyzed the mechanics of foot landing and the resultant impulse using a four-link chain model of the leg and torso. The joints are either completely rigid or infinitely compliant when the foot undergoes a rigid, inelastic collision with the ground ('Collision model' in Methods). The models at the two extremes of joint stiffness bound the experimental data, with the compliant model underestimating the measured fore-aft impulse while the stiff model overestimates it (*Figure 6c, d*, and *Figure 6—figure supplement 1*). This is expected because the muscle contraction needed for weight support and propulsion would induce non-zero but non-infinite stiffness at the joints. Although both models overestimate the dependence of the fore-aft impulse on foot speed, the slope of the compliant model is closest to the measurements (*Figure 6c*, *Figure 6—figure supplement 1*). The slope of measured speed-impulse data is $0.01 \pm 0.003$ ($p = 0.001$, *Table 3*), closer to compliant model than the stiff model, whose slopes are $0.0203 \pm 0.010$ ($p < 0.0001$) and $0.056 \pm 0.005$ ($p < 0.0001$), respectively. The measured fore-aft impulse for most steps was below 0.07 (whiskers extend to 1.5 times the interquartile range in *Figure 6*). The compliant model's predicted fore-aft impulses show good agreement with measurements when the impulse is below 0.07 (measured versus predicted in *Figure 6d*), and disagree only for the occasional steps when runners experience more severe fore-aft impulses. Unlike the compliant model, the stiff model consistently over-estimates the measured fore-aft impulse over its entire range. Thus, we propose that maintaining low joint stiffness at landing helps maintain low fore-aft impulses despite variations in touchdown foot speed.

**Table 1.** Correlation between landing probability and terrain unevenness.
Details of the ANCOVAs on the linear mixed models from *Equation 12* showing denominator degrees of freedom, F-values, and p-values from the dataset of stepping probabilities and terrain height statistics of 1515 recorded $p_{i,j}$ values for all subjects on uneven I and uneven II. Since the foot placement index $p_{i,j}$ values show very little variability (*Figure 5—figure supplement 3*), the model with the median terrain height was singular.

| Independent variable | DenDF | F-value | p-Value |
|---|---|---|---|
| IQR terrain height | 20.6 | 3.03 | 0.10 |

The online version of this article includes the following source data for table 1:

**Source data 1.** Subject-wise statistics of the terrain's height in heel-sized patches and the probability of stepping in that patch.

**Table 2.** Kinematic variables on different terrain types reported as mean ± SD, except for meander values which are reported as median ± interquartile range.

For each variable, we show details of the ANOVAs performed on the linear model in *Equation 10*, i.e., the F-value and p-value for the terrain factor. The denominator degrees of freedom for all ANOVAs was 16. Post-hoc comparisons are reported when the ANOVAs reached the significance bound of $\alpha = 0.05$.

| Variable | Flat | Uneven I | Uneven II | F-value | p-Value |
|---|---|---|---|---|---|
| Net metabolic rate (W/kg) | $13.1 \pm 0.5$ | $13.7 \pm 0.9$ | $13.7 \pm 0.8$ | 2.97 | 0.08 |
| Median step width (%LL) | $3.9 \pm 1.9$ | $4.1 \pm 1.5$ | $4.7 \pm 2.0$ | 4.53 | 0.03 |
| IQR step width (% LL) | $3.9 \pm 1.4$ | $4.3 \pm 0.9$ | $5.0 \pm 1.2$ | 3.65 | 0.05 |
| Mean step width (%LL) | $4.2 \pm 1.7$ | $4.7 \pm 1.6$ | $5.2 \pm 1.7$ | 8.69 | 0.003 |
| SD step width (% LL) | $2.8 \pm 0.8$ | $3.4 \pm 0.6$ | $3.6 \pm 0.6$ | 5.54 | 0.01 |
| Mean step length (%LL) | $128 \pm 6$ | $126 \pm 9$ | $125 \pm 9$ | 1.07 | 0.37 |
| SD step length (%LL) | $6 \pm 1$ | $7 \pm 4$ | $6 \pm 1$ | 0.64 | 0.54 |
| Mean meander ($\times 10^{-4}$) | $3.21 \pm 2.59$ | $3.97 \pm 1.65$ | $4.88 \pm 4.62$ | 1.48 | 0.25 |
| SD meander ($\times 10^{-4}$) | $0.67 \pm 0.53$ | $1.33 \pm 1.40$ | $1.27 \pm 2.78$ | 1.58 | 0.23 |
| Mean fwd. foot speed (froude num.) | $0.53 \pm 0.17$ | $0.36 \pm 0.10$ | $0.37 \pm 0.12$ | 13.08 | 0.0004 |
| SD fwd. foot speed (froude num.) | $0.17 \pm 0.05$ | $0.14 \pm 0.05$ | $0.18 \pm 0.07$ | 1.48 | 0.26 |
| Mean CoM speed (m/s) | $3.24 \pm 0.07$ | $3.21 \pm 0.07$ | $3.18 \pm 0.09$ | 2.32 | 0.13 |
| SD CoM speed (m/s) | $0.11 \pm 0.03$ | $0.13 \pm 0.04$ | $0.12 \pm 0.03$ | 2.00 | 0.17 |
| Mean touchdown leg length (%LL) | $120 \pm 5$ | $119 \pm 4$ | $119 \pm 4$ | 4.28 | 0.03 |
| SD touchdown leg length (%LL) | $1.1 \pm 0.7$ | $0.9 \pm 0.3$ | $1.3 \pm 1.2$ | 1.32 | 0.29 |
| Mean touchdown leg angle (rad) | $0.20 \pm 0.02$ | $0.20 \pm 0.02$ | $0.21 \pm 0.02$ | 3.90 | 0.04 |
| SD touchdown leg angle (rad) | $0.03 \pm 0.02$ | $0.02 \pm 0.003$ | $0.03 \pm 0.02$ | 2.10 | 0.15 |

The online version of this article includes the following source data for table 2:

**Source data 1.** Subject-wise, per-step data on foot and leg kinetics and kinematics.

## Leg retraction

Increased leg retraction rate results in reduced forward foot speed at touchdown, thereby altering the fore-aft impulse (*Karssen et al., 2015*; *Dhawale et al., 2019*). The mean non-dimensional forward foot speed at landing is terrain-dependent and lower by $0.17 \pm 0.04$ (p = 0.001) on uneven I compared to flat ground, and by $0.15 \pm 0.04$ (p = 0.002) on uneven II compared to flat ground (*Figure 7a*, *Table 2*). For the mean subject, these correspond to reductions in forward foot speed of $0.48 \pm 0.11$ m/s on uneven I and $0.42 \pm 0.11$ m/s on uneven II compared to flat ground.

We find that touchdown angle depends significantly but only weakly on forward foot speed at landing (p ≈ 0, slope = $0.07 \pm 0.01$ rad, *Table 3*). If the dimensionless forward foot speed at landing

**Table 3.** Details of the ANCOVAs performed on the linear model described in *Equation 11* showing the denominator degrees of freedom, F-value and p-value for the fixed terrain factor, and the estimated slopes $\beta_f$ for the fixed forward foot speed effect.

| Dependent variable | Factor | DenDF | F-value | p-Value | $\beta_f$ |
|---|---|---|---|---|---|
| Touchdown leg angle | Terrain | 193 | 1.48 | 0.23 | - |
| | Fwd. foot speed | 38 | 115.83 | <0.0001 | 0.07±0.01 rad |
| Fore-aft impulse | Terrain | 79 | 1.45 | 0.24 | - |
| | Fwd. foot speed | 78 | 12.83 | 0.001 | 0.01±0.003 |

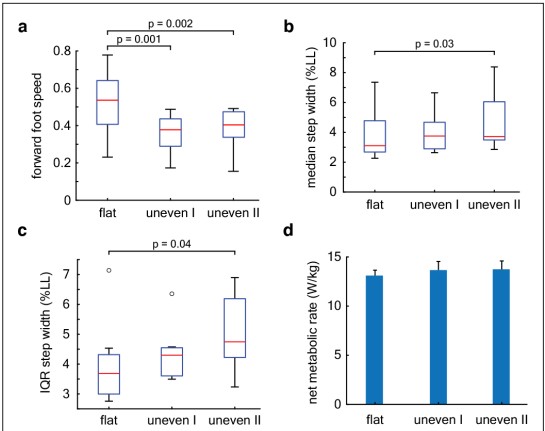

**Figure 7.** Energetics and stepping kinematics. (**a**) Box plot of the mean forward foot speed at landing (units of froude number). (**b**) Box plot of the median step width (normalized to leg length). (**c**) Box plot of the step width variability. Central red lines denote the median, boxes represent the interquartile range, whiskers extend to 1.5 times the quartile range, and open circles denote outliers. The distribution of step widths within a trial deviated from normality and hence we report the median and the interquartile range of the distribution for each trial (**Figure 7—figure supplement 1**), instead of the mean and standard deviation as is reported for all other variables. (**d**) Net metabolic rate normalized to subject mass. Whiskers represent standard deviation across the nine subjects. An ANOVA on the linear mixed model described in **Equation 10** was used to determine whether gait measures described above differed between terrain conditions with a significance threshold of 0.05.

The online version of this article includes the following source data and figure supplement(s) for figure 7:

**Source data 1.** Subject-wise, per-step data on step width.

**Figure supplement 1.** Subject-wise step width statistics.

**Figure supplement 2.** Representative respirometry data.

___

varied through its entire observed range from −0.2 to 1.1, it would result in a change in landing angle of 0.08 rad or 5°.

## Stepping kinematics

We find that the median non-dimensional step width is terrain dependent (**Figure 7b**, **Table 2**) and increased on uneven II versus flat ground by $0.004 \pm 0.001$ (p = 0.03). Step width variability, i.e., the interquartile range of step widths within a trial, is also terrain dependent (p = 0.05, **Figure 7c**, **Table 2**) and greater on uneven II versus level ground by $0.005 \pm 0.002$ (p = 0.04). For the mean subject, median step width increased by $4 \pm 1$ mm and the step width variability (IQR) increased by $6 \pm 2$ mm.

## Energetics

The approximately 5% increase in metabolic power consumption on the uneven terrain compared to flat we measured was not statistically significant ($p = 0.08$, **Figure 7d**, **Table 2**).

## Discussion

Our primary finding is that runners do not use visual information about terrain unevenness to guide their footsteps. In addition, the fore-aft collisions that they experience seem almost decoupled from the forward speed with which their foot lands on the ground. Based on the modeling estimate of collisional impulses and comparison with measurements, we propose that low joint stiffness underlie the regulation of fore-aft impulses, likely contributing to stability (**Dhawale et al., 2019**). Taken together, these results suggest that runners rely not on vision-based path planning, but on their body's passive mechanical response for remaining stable on undulating uneven terrain. Additionally, the changes in step-width kinematics on the uneven versus flat terrain may reflect sensory feedback mediated stepping strategies similar to those reported previously (**Seipel and Holmes, 2005**; **Seethapathi and Srinivasan, 2019**), but more work is needed to investigate whether the differences were the result of feedback control or simply the result of variability injected by the terrain's unevenness.

Measurements of fore-aft impulses have not been previously examined in the context of stability. A previous theoretical analysis hypothesized that reducing tangential collisions and maintaining low fore-aft impulses reduces the risk of falling by tumbling in the sagittal-plane (*Dhawale et al., 2019*). Our data are consistent with this model. We find that only $6 \pm 1\%$ of the forward momentum was lost in stance although the forward foot speed at landing varied by nearly 50%. This reduction in variability is surprising because, all else held the same, speed and impulse are expected to be linearly related. This suggests that the fore-aft impulse is tightly regulated by other means. By examining the role of leg joint compliance using model-based analyses of the data, we found that the measured fore-aft impulses were partly consistent with an idealized extreme of zero stiffness in the joints at the point of landing. However, joint stiffness in a real runner cannot be too small because it is needed to withstand the torques for weight support and propulsion. Thus, we propose that the low variability in fore-aft impulses arises from active regulation of joint stiffness.

Past studies on running birds (*Blum et al., 2014*; *Birn-Jeffery et al., 2014*) provide some hints on why leg compliance, and not foot speed, might be the preferred means to regulate fore-aft impulses. To deal with abrupt changes in terrain height, running birds regulate foot speed and leg retraction rates to maintain consistent leg forces and reduce discomfort or injury risk. Although our terrain has smoothly varying terrain and not the step-like blocks used in the bird studies, our runners may still have encountered sudden height changes because they did not precisely regulate their stepping pattern to avoid uneven terrain areas. Like the running birds, they may have regulated foot speed to mitigate discomfort and high forces. Thus, by employing leg compliance to reduce the fore-aft impulse, the runners could deal with stability independent of foot speed regulation for safety and comfort. However, caution is warranted when comparing our results with these past studies. The bird studies used SLIP models to interpret their findings, but such models are energy conserving and unaffected by slope variations that were part of our terrain design. Furthermore, the peak-to-peak height variation of our terrain was less than 6% of the leg length, unlike *Blum et al., 2014* and *Birn-Jeffery et al., 2014*, who used larger step-like obstacles of 10% leg length or more. For example, we see no change in the variability of the leg landing angle between flat and uneven terrain trials (*Table 2*), which is expected if leg landing angle responded to variations in terrain height (*Blum et al., 2014*; *Birn-Jeffery et al., 2014*). So large step-like obstacles probably induce different swing-leg control strategies compared with undulating terrain with smaller height variations.

We found variability in step-to-step kinematics that are largely consistent with previous studies on step-like terrain, but with some notable differences. Studies of running birds hypothesize that crouched postures could aid stability on uneven terrain (*Blum et al., 2011*; *Birn-Jeffery and Daley, 2012*), as do human-subject data from treadmill running (*Voloshina and Ferris, 2015*). We find a slight decrease in the virtual leg length at touchdown on the most uneven terrain compared to flat, but the difference was only around 1% of the leg length (*Table 2*), whose effect on stability would be negligible. We find higher leg retraction rates on uneven terrain, as also reported in running birds (*Birn-Jeffery and Daley, 2012*; *Blum et al., 2014*). Leg retraction has been hypothesized to improve running stability in the context of point-mass models by altering leg touchdown angle to aid stability (*Seyfarth et al., 2003*; *Blum et al., 2010*). However, we find only a weak dependence between leg retraction rate and leg touchdown angles. Human-subject treadmill experiments report that step width and step length variability increased by 27% and 26%, respectively, and mean step length or step width were the same for flat and uneven terrain (*Voloshina and Ferris, 2015*). Like those studies, we find 24% greater step width variability on uneven terrain compared to flat, but no significant changes in step length variability (*Figure 7b*, *Table 2*). We additionally find that the median step width increased on uneven terrain by 13%. The increase in median step width that we measure could be due to lateral stability challenges of running on relatively more complex terrain with smoothly varying slope and height variations in all directions.

Unlike treadmill running studies, we do not find a statistically significant increase in metabolic power consumption on uneven terrain versus flat ground, but the mean increase of around 5% is similar to *Voloshina and Ferris, 2015*. The acceleration and deceleration when subjects turn around during our overground trials could affect the metabolic energy expenditure. Therefore, caution is warranted in comparing the absolute value of our reported energetics data with other studies on treadmills or unidirectional running. But several aspects of the experimental design allow us to compare the respirometry data between the different terrain types. For every subject, we ensured that

the breath-by-breath respirometry data stabilized within the first 3 min and only used the stabilized value for further analyses ('Energetics' in Methods). If the transients had dominated the respirometry measurements, the measurements would not have stabilized (*Figure 7—figure supplement 2*). The use of the moving light bar on either side of the track ensured that the subjects maintained the same speed on all the terrain types. Moreover, the turnaround patches were designed to have the same terrain statistics (flat, uneven I, uneven II) as the rest of the track, thus ensuring that there were no abrupt terrain transitions. This allowed us to control for and mitigate the effects of the turnaround phases when comparing the results between the different terrain types.

We find no evidence that subjects used visual information from the terrain geometry to plan footsteps despite predicted advantages to stability (*Dhawale et al., 2019*). This finding differs from walking studies that highlight the role of vision in guiding step placement on natural, uneven terrain (*Matthis et al., 2018*; *Bonnen et al., 2021*). The stochastic stepping model was able to consistently find landing locations with lower unevenness than the human subjects, while matching the measured mean stepping statistics and even reducing step-to-step variability, thus showing that the absence of a foot placement strategy was not due to a lack of feasible landing locations. We speculate that foot placement strategies are used for obstacle avoidance (*Matthis and Fajen, 2014*) on more complex terrain while our terrains were designed to be continuously undulating and not have large, singular obstacles. While our data suggest that terrain-guided foot placement strategies are not required for stability on gently undulating terrain, it leaves open the possibility that there is a skill-learning component to such foot placement strategies which we could not measure since our volunteers were not experienced trail runners. Further experiments with runners of varying skill levels could test such a hypothesis.

## Conclusions

Footsteps were not directed toward flatter regions of the terrain despite predicted benefits to stability. Instead, we found evidence for a previously uncharacterized control strategy, namely that the body's stabilizing mechanical response due to low fore-aft impulses was used to mitigate the destabilizing effects of stepping on uneven areas. The limited need for visual attention may explain how runners could employ vision for other functional goals, such as planning a path around large obstacles, or in an evolutionary context, tracking footprints to hunt prey on uneven terrain without falling. Whether other animals employ similar strategies on uneven terrain is presently unknown but data from galloping dogs show that they do not alter their gait on uneven terrain (*Wilshin et al., 2020*), thus suggesting that other adept runners potentially employ similar principles for stability. We propose that our results could translate to new strategies for reducing the real-time image processing burden in robotic systems, and could also help in training trail runners by emphasizing limber joints when dealing with uneven terrain.

## Acknowledgements

Human Frontier Science Program and Wellcome Trust-DBT Alliance for funding.

## Additional information

### Funding

| Funder | Grant reference number | Author |
|---|---|---|
| Human Frontier Science Program | RGY0091/2013 | Madhusudhan Venkadesan |
| The Wellcome Trust DBT India Alliance | | Madhusudhan Venkadesan |

The funders had no role in study design, data collection and interpretation, or the decision to submit the work for publication. For the purpose of Open Access, the authors have applied a CC BY public copyright license to any Author Accepted Manuscript version arising from this submission.

## Author contributions
Nihav Dhawale, Data curation, Software, Formal analysis, Validation, Investigation, Visualization, Methodology, Writing – original draft, Writing – review and editing; Madhusudhan Venkadesan, Conceptualization, Resources, Data curation, Formal analysis, Supervision, Funding acquisition, Validation, Investigation, Visualization, Methodology, Writing – original draft, Project administration, Writing – review and editing

## Author ORCIDs
Nihav Dhawale ⦿ http://orcid.org/0000-0002-5193-9064
Madhusudhan Venkadesan ⦿ http://orcid.org/0000-0001-5754-7478

## Ethics
Human subjects: The study was approved by the Institute Ethics Committee (Human Studies) of the National Centre for Biological Sciences, Bengaluru, India (TFR:NCB:15\_IBSC/2012), where the experiments were conducted. Informed consent was obtained by the experimenter N. Dhawale and M. Venkadesan, who are the authors of this manuscript. The procedure followed for seeking informed consent followed the steps that were approved by the Ethics Committee mentioned above.

## Decision letter and Author response
Decision letter https://doi.org/10.7554/eLife.67177.sa1
Author response https://doi.org/10.7554/eLife.67177.sa2

# Additional files

## Supplementary files
• MDAR checklist

## Data availability
All data points plotted are in either the main text, the figure supplements, or source data attached to figures and tables.

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
