## [Editor Report]

This paper presents fundamental evidence for the control mechanisms used by running humans to maintain stability while running on naturalistically uneven terrain. The authors use a creative and compelling combination of experiments and modeling to analyze running on terrain with mildly stochastic undulating roughness, a condition that resembles natural terrain conditions, such as trail running. The findings suggest that humans use open-loop, intrinsically stable strategies to run on this terrain, and not visually guided foot placement, making an important contribution to understanding the context-dependent role of vision in human locomotion.

---

## [Decision Letter]

**Decision letter after peer review:**

Thank you for submitting your article "How human runners regulate footsteps on uneven terrain" for consideration by *eLife*. Your article has been reviewed by 2 peer reviewers, and the evaluation has been overseen by a Reviewing Editor and Richard Ivry as the Senior Editor. The following individuals involved in the review of your submission have agreed to reveal their identity: Andrew A Biewener (Reviewer #2).

Essential revisions:

This paper presents findings from a novel experimental study of the dynamics of human overground running on uneven terrain. The terrain used in the experiments has mildly stochastic undulating roughness, a condition that closely resembles many natural terrain conditions, such as trail running. The most compelling and innovative contribution of the current work is the investigation of how foot placement by the human subjects compares to model-based predictions of 'directed' and 'blind' foot placement schemes. The model-based interpretation of the experimental dataset provides clear evidence that humans do not use visually guided foot placement in these conditions. Instead, subjects appear to use open-loop intrinsically stable strategies to run in this specific terrain context. This contrasts with some findings of human locomotion at slower walking speeds and navigating terrain with large obstacles. Experimental data on the mechanics and sensorimotor control of running in uneven terrain remain sparse, and therefore the current findings make an important contribution toward understanding the context-dependent role of vision in navigating uneven terrain.

The editor and expert reviewers agreed that the paper has the potential to make a novel and substantive contribution; however, they also raised important concerns that need to be addressed through revisions to the text, and potential re-analysis of the metabolic cost data. The two most essential points are summarized below:

1) The current text does not accurately describe the experimental conditions in the prior work by Voloshina and Ferris 2015, and this leads to some overstatement of the novelty of the undulating terrain conditions. Voloshina and Ferris investigated human running over stochastically uneven terrain with height variability smaller than the human foot, which resulted in landing conditions with slope variation. The authors should revise the text to acknowledge the similarity of these experimental conditions and make more direct and thorough comparisons between these studies in the paper.

2) The metabolic cost measures are confounded by turning at each end of the runway. The authors need to address this limitation more thoroughly in the text. It is nonetheless reassuring that the increase in metabolic cost is comparable in magnitude to that found by Voloshina and Ferris 2015 on an uneven terrain treadmill, and replication of similar energetic findings in an overground setting is an interesting outcome.

---

## [Author Response]

Essential revisions:This paper presents findings from a novel experimental study of the dynamics of human overground running on uneven terrain. The terrain used in the experiments has mildly stochastic undulating roughness, a condition that closely resembles many natural terrain conditions, such as trail running. The most compelling and innovative contribution of the current work is the investigation of how foot placement by the human subjects compares to model-based predictions of 'directed' and 'blind' foot placement schemes. The model-based interpretation of the experimental dataset provides clear evidence that humans do not use visually guided foot placement in these conditions. Instead, subjects appear to use open-loop intrinsically stable strategies to run in this specific terrain context. This contrasts with some findings of human locomotion at slower walking speeds and navigating terrain with large obstacles. Experimental data on the mechanics and sensorimotor control of running in uneven terrain remain sparse, and therefore the current findings make an important contribution toward understanding the context-dependent role of vision in navigating uneven terrain.

Thank you for the succinct and accurate summary of the main contributions of our work. We appreciate the thoughtful reviews and the opportunity to submit a revised manuscript. As accurately summarized in the reviews, our results on uneven terrain running provide clear evidence that humans do not visually guide their footfalls towards level areas and instead likely rely on open-loop intrinsically stable strategies. Our findings are on terrain with unevenness similar to natural trails and show interesting differences from previous work on walking or running in areas with large obstacles.

The reviewers were appreciative of our work but raised critical points, responding to which has helped improve the manuscript significantly. The revisions include new analyses and methodological improvements, and extensive revisions to the text throughout the manuscript.

This response document is organized to first present our responses to the points that are raised in the editor’s summary, followed by responses to the referees. The revised submission includes the revised manuscript, and a separate annotated version highlighting the revisions.

The editor and expert reviewers agreed that the paper has the potential to make a novel and substantive contribution; however, they also raised important concerns that need to be addressed through revisions to the text, and potential re-analysis of the metabolic cost data. The two most essential points are summarized below:

Thank you for the encouragement and the comments to help improve our manuscript. We have responded to the two major points raised in the editor’s summary through: (i) additional analyses to better relate our work to that of Voloshina and Ferris (2015), and (ii) a more thorough discussion of the potential limitations associated with the energetics measurement in the back-and-forth running protocol, and how our experimental design controls for this limitation.

1) The current text does not accurately describe the experimental conditions in the prior work by Voloshina and Ferris 2015, and this leads to some overstatement of the novelty of the undulating terrain conditions. Voloshina and Ferris investigated human running over stochastically uneven terrain with height variability smaller than the human foot, which resulted in landing conditions with slope variation. The authors should revise the text to acknowledge the similarity of these experimental conditions and make more direct and thorough comparisons between these studies in the paper.

Thank you for raising this important discussion on the relationship between our terrain design and that of Voloshina and Ferris (2015). The previous study by Voloshina and Ferris (2015) used a step-like pattern of blocks, but the blocks were themselves narrow enough that the foot probably spanned more than one fore-aft block. As the editor and referees correctly state, the hind and forefoot would be at different heights and the foot would not be horizontal during foot flat. This feature is similar to our study but there are also some noteworthy differences in how the terrain affects the body in our study. We have now revised the text to note the similarities and differences between our design and that of previous studies.

The similarity between our study and the past design using blocks is that both lead to variation in the foot angle at foot flat. In Voloshina and Ferris (2015), the blocks were of three different heights (say, A, B, and C), which leads to six possible height difference pairings (AB, BA, AC, CA, BC, CB). But in our study, the variation in slope is continuously graded, which would allow for more variation in the foot flat angle. Moreover, it is not only the foot angle that affects whole body dynamics, but the force direction from the ground also matters. In this regard, our design may differ from the block design, particularly during initial contact and push-off when only a small region of the foot makes contact with the ground. During push-off or foot landing, the slope induced by the block design may not manifest in the ground forces because just the rear or forefoot region would be pushing on the ground. But for our designed terrain, like natural undulating terrain, the ground would be sloped even when a small portion of the foot is in contact with it and would lead to ground forces that are different from flat ground.

Although further study is needed to systematically compare the two designs, the similarity in the measured energetics and kinematics between our study and those of Voloshina and Ferris (2015) suggests that the block design probably has a similar effect as our terrain design. Our study builds on the past work by establishing similarity in energetics and kinematics, and reports new findings on the foot placement strategy that is used by the runners. The use of an overground, continuously undulating terrain enabled us to conduct the foot placement analysis, which relied on the continuously varying unevenness of the terrain to assess whether the stepping pattern was affected by terrain unevenness. Furthermore, the use of an overground running track allowed the runners to scan the oncoming terrain over several foot steps, thus allowing some generalization of our findings to natural terrain.

In the last paragraph of the revised Introduction, we motivate the need for a study using undulating terrain and present a comparison of our terrain design with past work, including Voloshina and Ferris (2015). Further revisions have been made throughout the manuscript to carefully differentiate between step-like terrain, uneven terrain made of rectangular blocks that could induce variations in the foot angle, and smoothly undulating terrain.

2) The metabolic cost measures are confounded by turning at each end of the runway. The authors need to address this limitation more thoroughly in the text. It is nonetheless reassuring that the increase in metabolic cost is comparable in magnitude to that found by Voloshina and Ferris 2015 on an uneven terrain treadmill, and replication of similar energetic findings in an overground setting is an interesting outcome.

The review has identified an important consideration that went into our study design, including the choice of an experimental space where we could conduct this study. The main aspects of our study design to reduce the impact of the turn at each end of the track are the following:

1. A moving light bar was used to control speed so that subjects maintained a steady speed for much of the track. Of the 24m track, the experimenters verified that the subjects maintained a steady speed and stayed within moving light bar for the 21.6m long straight portion of the track.

2. A wider turn-around terrain patch that matched the terrain characteristics of the rest of the track was designed so that subjects did not step out of the terrain. Thus, we avoided any abrupt terrain transitions during the run.

3. The comparison between terrain types was the focus, and all three involved the transients associated with the turn at the end of the track. Therefore, we are able to compare between the different terrain types used in our study, despite the limitations in comparing our measurements to previous treadmill studies.

The relevant portions in the revised manuscript appear in the Methods and Discussion sections, and reproduced below:

Methods section 2.1.5:

“Of the total track length of 24m, a 1.2m turnaround segment was designed at each end to facilitate the subjects to reverse their running direction without stepping off the track. These end segments were 1m wide, which was broader than the rest of the track that was only 0.6m wide. The runners would reach the end of the track and turn around promptly. Guiding light bars that controlled the running speed would be half “absorbed” into the end before reversing direction, which allowed for sufficient time for the subjects to turn around while still maintaining the same average speed. The subjects were given, and took, around 0.5s to turn around. The subjects ran at a steady speed within the capture volume that covers the middle 10m of the track (see results for details). The cameras could not capture the ends of the track but the experimenters observed that the subjects stayed within the moving light bar through the 21.6m long straight portion of the track. The experimental protocol used in this study was tuned through pilot trials involving the authors of this manuscript and 2 initial subjects. The data from these pilot trial subjects are not part of the reported results in this manuscript.”

Discussion section:

“Unlike treadmill running studies, we do not find a statistically significant increase in metabolic power consumption on uneven terrain versus flat ground, but the mean increase of around 5% is similar to Voloshina and Ferris (2015). The acceleration and deceleration when subjects turn around during our overground trials could affect the metabolic energy expenditure. Therefore, caution is warranted in comparing the absolute value of our reported energetics data with other studies on treadmills or unidirectional running. But several aspects of the experimental design allow us to compare the respirometry data between the different terrain types. For every subject, we ensured that the breath-by-breath respirometry data stabilized within the first 3 minutes and only used the stabilized value for further analyses (Methods 2.1.4). If the transients had dominated the respirometry measurements, the measurements would not have stabilized (Figure 7—figure supplement 2). The use of the moving light bar on either side of the track ensured that the subjects maintained the same speed on all the terrain types. Moreover, the turnaround patches were designed to have the same terrain statistics (flat, uneven I, uneven II) as rest of the track, thus ensuring that there were no abrupt terrain transitions. This allowed us to control for and mitigate the effects of the turnaround phases when comparing the results between the different terrain types.”